# Uncertain Programming Model for the Cross-Border Multimodal Container Transport System Based on Inland Ports

**Junchi Ma, Xifu Wang, Kai Yang * and Lijun Jiang**

School of Traffic and Transportation, Beijing Jiaotong University, Beijing 100044, China
* Correspondence: kaiyang@bjtu.edu.cn

**Abstract:** The importance of inland ports in promoting current cross-border trade is increasingly recognized. In this work, we aim to design the entire network for the cross-border multimodal container transport system based on inland ports. Unlike previous studies, we consider strong uncertainty in cross-border transportation demand to be caused by a variety of realistic factors such as the global economic situation, trade policies among countries, and global epidemics, etc. To handle the demand uncertainty, we develop an uncertain programming model for the considered cross-border multimodal container transportation network design problem to minimize the expectation of the total costs, including carbon emissions, by imposing two types of chance constraints for capacity limitations. Under mild assumptions, we further convert the proposed uncertain model into its equivalent deterministic one, which can be solved by off-the-shelf solvers such as CPLEX, Gurobi, and Lingo. Finally, we illustrate the applicability of the proposed model by taking the Huaihai Economic Zone-Europe multimodal container transport system as a real-world case study. The computational results provide valuable suggestions and policy guidance regarding four issues: the inland port locations, the transportation route choices, the strategies for reducing the total cost, and the schemes for improving network performance against uncertain demand.

**Keywords:** uncertainty theory; inland port; multimodal transport system; uncertainty programming; China-Europe Railway Express



## 1. Introduction

Inland ports are the key nodes for inland cities to connect to foreign countries. In Europe, the inland ports are typically recognized as an extension of seaport functions in inland areas [1], whereas in China, the inland ports are more frequently recognized as independent logistics hubs, relying on multiple transportation modes with the China-Europe Railway Express (CRE) as the core to achieve direct channel connections with foreign hubs [2]. In contrast to conventional logistics parks, the inland ports in China are marked by larger handling capacities, more transportation modes, the ability to open CRE, customs functions in inland cities, and clear economies of scale [2]. According to the China Inland Port Development Report [3], the number of inland ports in China has risen to 203 by the end of 2021. From this, the function of inland ports in promoting international trade flow is gaining prominence.

With the implementation of the Belt and Road Initiative, cross-border trade in Chinese cities has seen rapid growth [4]. From 2013 to 2021, China's cumulative import and export amount of products was CNY 262.3 trillion, with an average annual growth rate of 5.4% [4]. Likewise, the progress of the CRE is also outstanding [5]. The CRE has already operated 49,000 trains, transported 4,432,000 TEUs of goods, and reached 180 cities in 23 European countries by the end of 2021 [5]. With the logistics service network covering the entire territory of Asia and Europe, CRE is becoming an international public logistics product widely recognized by countries along the route. Therefore, the inland ports and CRE trains are both important components of China's cross-border trade logistics network, especially under the Belt and Road Initiative [3,5,6].



Even though the number of Chinese inland ports and the volume of cross-border trade have grown rapidly in recent years, it should be noted that there are still issues with the network layout of the China-Europe multimodal container transport system. On the one hand, some city nodes are over-subsidizing the operation of CRE and adopting low-priced strategies to compete with maritime transport for cargo sources, which distorts the market order to a certain extent [6]. On the other hand, some regions in the network of China have demonstrated the problem of over-intensive inland ports, which has led to increased competition among inland ports and the waste of transport and train resources [7]. With this concern in mind, we concentrate on the network layout for the cross-border multimodal container transport system based on inland ports between China and Europe.

Generally speaking, the transportation demand between inland cities and foreign hubs is influenced by multiple factors, including the global economic situation, trade policies between countries, exchange rate fluctuations, global epidemics, sudden natural disasters, etc., all of which will bring uncertainty. Based on the aforementioned background, this study addresses the following research questions:

- How to design the entire network for the cross-border multimodal container transport system based on inland ports under uncertain demand conditions?
- How to identify the impact of different factors on the optimal network structure?
- What strategies can we propose to improve network performance against uncertain demand?

In order to answer the above research questions, this paper introduces the cross-border multimodal container transportation network design problem. Given the importance of carbon neutral policies, this paper especially considers the cost of carbon emissions, which can be expressed as the product of the volume of carbon emissions and the amount of carbon tax, and analyses the situations when the carbon tax changes. It is worth noting that the assumption of deterministic demand often becomes restrictive in the real-world cross-border multimodal container transport system. By contrast, this paper employs the uncertain variables to describe the uncertainty associated with demand via the experiences and judgments of field experts. Furthermore, this paper proposes an uncertain programming model for the cross-border multimodal container transportation network design problem to minimize the expected total costs by imposing two types of chance constraints for capacity limitations. To solve the proposed model efficiently, this paper also adopts uncertainty theory to convert the uncertain model into an equivalent crisp one, which can be directly handled by off-the-shelf solvers such as CPLEX, Gurobi, and Lingo. Finally, this paper takes the Huaihai Economic Zone-Europe multimodal container transport system as a case study and presents several valuable suggestions and policy guidance with strong theoretical and practical implications.

This paper is structured as follows: Section 2 provides an overview of the current literature on inland ports, multimodal container transport systems, and uncertain theory. Section 3 develops the deterministic and uncertain programming models for the cross-border multimodal container transportation network design problem. Section 4 investigates a case study of the Huaihai Economic Zone-Europe multimodal container system. Section 5 analyzes the computational results and proposes policy guidance. Section 6 gives the main conclusions and makes recommendations for further research.

## 2. Literature Review

### 2.1. Inland Port and Its Role in Cross-Border Trade

The definition of the inland port is widely accepted as an inland intermodal terminal directly connected to seaport(s) with high-capacity transport mean(s), where customers can leave/pick up their standardized units as if directly to a seaport based on Leveque and Roso [8]. There have been a number of studies focusing on seaports. For example, Bernacki and Lis [9] investigated the future evolution of port systems, considering the case of Poland and the Rhine-Scheldt Delta region. Szaruga et al. [10] presented the synchronization of economic cycles of GDP and oil product cargo volumes in major Polish seaports. Kotowska

et al. [11] noticed that inland shipping to serve the hinterland is a big challenge for seaport authorities. Focusing on inland ports, Witte et al. [12] presented a systematic and integrated review of inland port studies, covering 80 international peer-reviewed academic journal papers on inland port development between 1992 and 2017, suggesting that the inland port has become a widely accepted concept. Wiegmans et al. [1] focused attention on the characteristics of inland waterway ports in a European context, proving that the definition of inland ports has been enriched beyond just the extension of seaports. Wang et al. [13] investigated the impact of inland port development in China on the promotion of bilateral trade flows between China and South Korea, suggesting that inland port development will increase cross-border trade volume. Monios and Wang [14] studied the spatial and institutional characteristics of inland port development in China, providing inland port development suggestions in a new geographical context. Xie et al. [15] suggested that China's foreign trade needs new momentum and that it is important to optimize the logistic network under the Belt and Road Initiative.

Despite years of research on inland ports, there is still a lack of solutions for the optimal layout of inland ports in the cross-border transport system. Additionally, there hasn't been much research done on the over-concentration of inland ports. To fill the research gap, this study develops a mathematical model to design the entire network by locating the inland ports.

### 2.2. Multimodal Container Transport System

There have been a number of models focusing on multimodal container transport systems that consider different indicators [16–22]. For instance, Jiang et al. [16] established a theoretical basis for the related exploration of multimodal container transportation. Corman et al. [17] presented a study on the influence and sensitivity of different model parameters, in order to analyze the implications on strategic decisions, fostering a target modal share for freight transportation. In order to describe the optimal organization problem, an optimization model based on dynamic programming was presented and was satisfied with reality constraints [18]. Hryhorak et al. [19] analyzed the dynamics and structure of freight transport in Ukraine. Containerized commodity transportation schemes are highly efficient for the majority of transcontinental and long-distance deliveries optimizing costs, and time quality of transport operations based on exact forecasting of container turnover. Zhang et al. [20] introduced a modeling approach for the optimization of terminal networks, taking into account the costs of $CO_2$ emissions and economies of terminal scale. Zehendner et al. [21] proposed a mixed integer linear programming model, based on a network flow representation of the terminal, to determine the number of appointments to accept per time slot and an allocation of internal resources minimizing service times of trains and barges simultaneously. Wei and Dong [22] proposed a cross-border logistics network based on inland ports and established a two-objective model in order to optimize freight cost and transport time at the same time.

The modeling of multimodal container transport systems has been the subject of a sizable body of research [16–21], but the cross-border multimodal container transport systems based on inland ports have received less attention. By combining the amount of carbon emissions with the carbon tax, this study also proposes a more precise method for estimating the cost of carbon emissions.

### 2.3. Application of Uncertainty Theory

Liu [23] creatively provided a self-contained, comprehensive, and up-to-date presentation of uncertainty theory. Liu [23–25] showed a more comprehensive understanding of uncertain theory, including uncertain programming, uncertain risk analysis, uncertain reliability analysis, uncertain process, uncertain calculus, uncertain differential equation, uncertain logic, uncertain entailment, and uncertain inference. With the development of uncertainty theory, there is a growing literature on the applications of uncertainty theory. For example, Gu and Zhu [26] investigated a new type of optimal control problem governed by

a parabolic uncertain partial differential equation in which the expected value criterion was adopted as the objective function. Shen and Zhu [27] employed uncertain programming to deal with the job shop scheduling problem with uncertain processing time and cost. Zhang and Peng [28] proposed the concepts of the expected shortest route, $\alpha$-shortest route, and distribution shortest route in the Chinese postman problem based on uncertainty theory. Ke et al. [29] proposed uncertain random multilevel programming for modeling decentralized decision-making problems with uncertain random parameters. Yang et al. [30] built a multi-period uncertain workforce planning model with job satisfaction level, with uncertainty in labor demands and operation costs.

However, since uncertainty theory is a relatively new methodology, there are not many applications for it in cross-border logistics. This work represents the first attempt to integrate uncertainty theory into the cross-border multimodal container transport system based on inland ports. By employing an uncertain programming approach to tackle the considered problem, the findings are more applicable to the practical cross-border logistics process.

## 3. Problem Formulations

In this study, we introduce the cross-border multimodal container transportation network design problem, which is to locate the inland ports and choose the transportation modes between nodes with minimum total costs. As displayed in Figure 1, we have classified the total costs into five types in this paper. The network optimization model for the case of deterministic demand is analyzed first. On this premise, the investigation into the uncertain demand case is conducted, and the problem is converted into a crisp model, leading to solutions for the inland port selection, transport route selection, and minimum cost for each case.

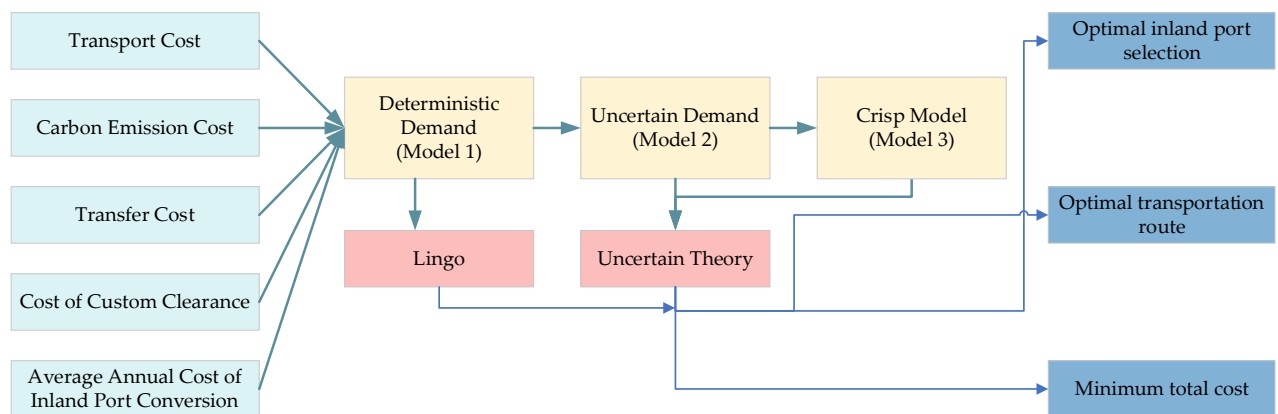

**Figure 1.** A schematic diagram of the stated work.

Based on the background described in Section 1, the network for the cross-border multimodal container transport system based on inland ports is illustrated in Figure 2. More specifically, the considered network contains five types of nodes: origin city, domestic logistics park, domestic inland port, domestic seaport, and foreign hub, and the inland port is formed by the expansion of the logistics park. The transportation mode from origin cities to logistics parks/inland ports, or seaports is road. The transportation modes between logistics parks/inland ports and seaports include road, railway, and inland waterways. The transportation mode between inland ports and foreign hubs is CRE. The transportation mode between seaports and foreign hubs is shipping. In this paper, we only consider the one-way transportation process from domestic cities to foreign hubs.

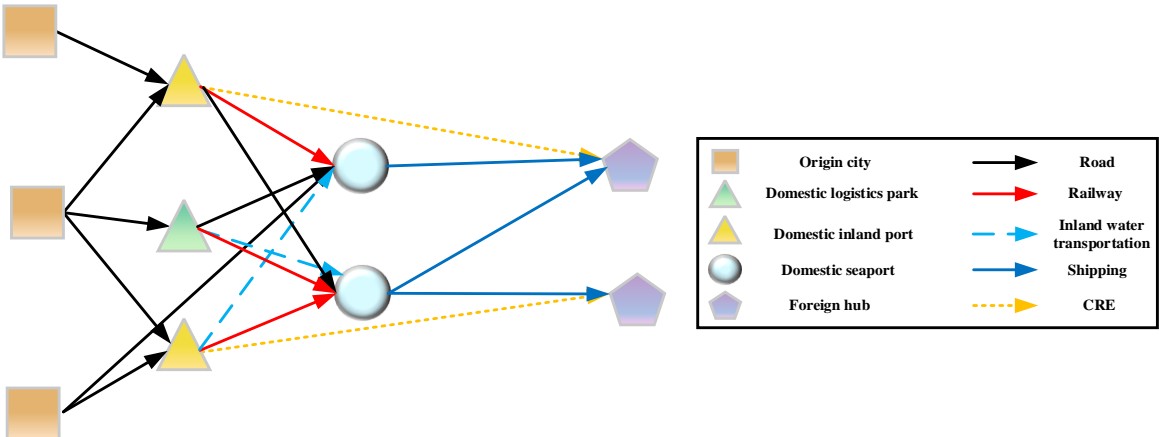

**Figure 2.** The network studied in this paper.

### 3.1. Definition of Symbols, Parameters, and Decision Variables

3.1.1. Symbols and Parameters

To formally characterize the problem, we first list all symbols and parameters used in the problem formulation, as shown in Table 1.

**Table 1.** Symbols and parameters used in this problem.

| Notations | Detailed Definition |
| --- | --- |
| $I$ | The set of all origin cities. |
| $J$ | The set of all foreign hubs. |
| $K$ | The set of all domestic logistics parks. |
| $L$ | The set of all domestic seaports. |
| $C_{ab}^m$ | The transport costs per TEU cargo from node $a$ to node $b$ by transport mode $m$ (yuan/TEU). |
| $D_{ab}^m$ | The transport distance from node $a$ to node $b$ by transport mode $m$ (km). |
| $m$ | The transport mode: $m = 1$ refers to road, $m = 2$ refers to railway, $m = 3$ refers to inland waterway, $m = 4$ refers to shipping, $m = 5$ refers to CRE. |
| $c^m$ | The transport costs per kilometer per TEU cargo by transport mode $m$ (yuan/km·TEU). |
| $c^m$ | The carbon emissions per kilometer per TEU cargo by transport mode $m$ (tonne/km·TEU). |
| $W_{m1m2}$ | The handling costs per TEU cargo converted from transport mode $m_1$ to $m_2$ (yuan/TEU). |
| $T_{m1m2}$ | The waiting time of cargo converted from transport mode $m_1$ to $m_2$ (day). |
| $H$ | The container occupancy cost per day per TEU cargo (yuan/TEU·day). |
| $G_l$ | The customs clearance costs per TEU cargo from seaport $l$ (yuan/TEU). |
| $G_k$ | The customs clearance costs per TEU cargo from inland port $k$ (yuan/TEU). |
| $F$ | The carbon tax per tonne of carbon emissions (yuan/tonne). |
| $Z_k$ | The average annual construction cost of converting node $k$ to an inland port (10,000 yuan). |
| $B$ | The average annual investment limit in the conversion of inland ports (10,000 yuan). |
| $Q_L$ | The maximum annual handling capacity of the domestic seaport $l$ (10,000 TEU). |
| $Q_k^1$ | The maximum annual handling capacity when node $k$ is not expanded into inland port (10,000 TEU). |
| $Q_k^2$ | The maximum annual handling capacity when node $k$ is expanded into inland port (10,000 TEU). |
| $q_{ij}$ | The annual transport demand from origin city $i$ to foreign hub $j$ (10,000 TEU). |

3.1.2. Decision Variables

In this problem, the decision variables concern the selection of inland ports and the transportation route from each origin city to each foreign hub. Considering different kinds of routes from origin city to foreign hub, we get six types of decision variables in total, as described below:

$$Y^1_{iklj} = \begin{cases} 1, & \text{if route } i \overset{\text{road}}{\to} k \overset{\text{road}}{\to} l \overset{\text{shipping}}{\to} j \text{ is selected} \\ 0, & \text{otherwise} \end{cases}$$

$$Y^2_{iklj} = \begin{cases} 1, & \text{if route } i \overset{\text{road}}{\to} k \overset{\text{railway}}{\to} l \overset{\text{shipping}}{\to} j \text{ is selected} \\ 0, & \text{otherwise} \end{cases}$$

$$Y^3_{iklj} = \begin{cases} 1, & \text{if route } i \overset{\text{road}}{\to} k \overset{\text{inland water}}{\to} l \overset{\text{shipping}}{\to} j \text{ is selected} \\ 0, & \text{otherwise} \end{cases}$$

$$Y_{ilj} = \begin{cases} 1, & \text{if route } i \overset{\text{road}}{\to} l \overset{\text{shipping}}{\to} j \text{ is selected} \\ 0, & \text{otherwise} \end{cases}$$

$$Y_{ikj} = \begin{cases} 1, & \text{if route } i \overset{\text{road}}{\to} k \overset{\text{CRE}}{\to} j \text{ is selected} \\ 0, & \text{otherwise} \end{cases}$$

$$X_k = \begin{cases} 1, & \text{if node } k \text{ is converted to an inland port} \\ 0, & \text{otherwise} \end{cases}$$

*3.2. Deterministic Model*

3.2.1. Calculation of Carbon Emissions

There are a number of approaches that can be used to estimate carbon emissions. Essentially, carbon emissions from the transportation process are affected by two primary elements: the kind of vehicle used (which is also regarded as the transportation method in this study) and the kind of energy used to power it. In this research, based on the formula for unit transport $CO_2$ emissions from the *IPCC Guidelines for National Greenhouse Gas Inventories (2006)* issued by the United Nations [31], we get the following formula to compute transportation carbon emissions in this study.

$$e^m = E^m \cdot H \cdot P \cdot \lambda \tag{1}$$

in which, $e^m$ represents the carbon emissions per unit distance per TEU transported by transportation mode $m$ (tonne). $E^m$ denotes the combined energy consumption of standard coal used per unit of transport workload for transport mode $m$ (kg). $H$ represents the calorific value of standard coal (GJ/kg). $P$ represents the emission factor of $CO_2$ (kg/GJ). $\lambda$ denotes the carbon oxidation factor. According to the IPCC database, $P$ is 25.8 if the fuel used is coking coal. In China, the calorific value of standard coal is 7000 kcal/kg, which is 29.3067 GJ/t in international units. The parameter $\lambda$ is usually taken as 1.

3.2.2. Assumptions

**Assumption 1.** The transportation demand from the domestic cities to the foreign hubs is determined.

**Assumption 2.** For the identified domestic cities and foreign hubs, only one transportation route can be used to meet transportation demand, and for a predetermined route, only one transportation mode can be taken between two adjacent points, which is consistent with the actual transportation process.

**Assumption 3.** Only inland ports have the necessary conditions for the opening of CRE trains, i.e., they have customs functions, a large handling capacity, and multimodal transportation.

**Assumption 4.** For transfer costs, only the costs associated with changing the mode of transportation are considered, and the transfer cost differences caused by infrastructures at different nodes are not taken into account.

3.2.3. Mathematical Formulation with Deterministic Demand

The objective of the optimization model is to minimize the total cost. Based on the notations and assumptions made before, we build the objective functions as in Equation (2). The objective function minimizes the total cost, which is composed of five

terms. $C_1$ represents the transport cost considering different routes and different transportation modes. $C_2$ represents the carbon emission cost, which corresponds to the volume of goods, the route and transportation mode option, and the carbon tax. $C_3$ represents the transfer cost, including the transfer handling cost and container occupancy cost, between different transportation modes. $C_4$ represents the cost of customs clearance and is related to whether the clearance point is an inland port or a seaport. $C_5$ represents the average annual cost of inland port conversion.

$$\min C_{total} = C_1(q_{ij}) + C_2(q_{ij}) + C_3(q_{ij}) + C_4(q_{ij}) + C_5$$

$$C_1(q_{ij}) = \sum_{i \in I} \sum_{j \in J} \sum_{k \in K} \sum_{l \in L} \sum_{m=1}^{3} q_{ij}[Y_{iklj}^m(C_{ik}^1 + C_{kl}^s + C_{lj}^4) + Y_{ilj}(C_{il}^1 + C_{lj}^4) + Y_{ikj}(C_{ik}^1 + C_{kj}^5)]$$

$$= \sum_{i \in I} \sum_{j \in J} \sum_{k \in K} \sum_{l \in L} \sum_{m=1}^{3} q_{ij}[Y_{iklj}^m(D_{ik}^1 c^1 + D_{kl}^m c^m + D_{lj}^4 c^4) + Y_{ilj}(D_{il}^1 c^1 + D_{lj}^4 c^5) + Y_{ikj}(D_{ik}^1 c^1 + D_{kj}^5 c^5)]$$

$$C_2(q_{ij}) = \sum_{i \in I} \sum_{j \in J} \sum_{k \in K} \sum_{l \in L} \sum_{m=1}^{3} q_{ij}F[Y_{iklj}^m(D_{ik}^1 e^1 + D_{kl}^m e^m + D_{lj}^4 e^4) + Y_{ilj}(D_{il}^1 e^1 + D_{lj}^4 e^5) + Y_{ikj}(D_{ik}^1 e^1 + D_{kj}^5 e^5)] \quad (2)$$

$$C_3(q_{ij}) = \sum_{i \in I} \sum_{j \in J} \sum_{k \in K} \sum_{l \in L} \sum_{m=1}^{3} q_{ij}[Y_{iklj}^m(W_{1m} + T_{1m}H + W_{m4} + T_{m4}H) + Y_{ilj}(W_{14} + T_{14}H) + Y_{ikj}(W_{15} + T_{15}H)]$$

$$C_4(q_{ij}) = \sum_{i \in I} \sum_{j \in J} \sum_{k \in K} \sum_{l \in L} \sum_{m=1}^{3} q_{ij}[(Y_{iklj}^m + Y_{ilj})G_l + Y_{ikj}G_k]$$

$$C_5 = \sum_{k \in K} Z_k X_k$$

We next present some basic constraints that the network design must satisfy, as described below. Constraint (3) ensures that there is only one route from the origin city $i$ to the foreign hub $j$. Constraint (4) ensures that only when the logistics park is upgraded to an inland port can a CRE train be opened at this node. Constraints (5) and (6) are capacity constraints for logistics parks, inland ports, and seaports. Constraint (7) ensures that the average annual total investment in inland ports is less than the investment limit. Constraints (8) and (9) are logical constraints on the decision variables.

$$\sum_{k \in K} \sum_{l \in L} Y_{iklj}^1 + Y_{iklj}^2 + Y_{iklj}^3 + Y_{ilj} + Y_{ikj} = 1, \forall i \in I, j \in J \quad (3)$$

$$\text{sign}\left(\sum_{i \in I} \sum_{j \in J} Y_{ikj}\right) = X_k, \forall k \in K \quad (4)$$

$$\sum_{i \in I} \sum_{j \in J} \sum_{l \in L} q_{ij}(Y_{iklj}^1 + Y_{iklj}^2 + Y_{iklj}^3 + Y_{ikj}) \leq Q_k^1 + X_k Q_k^2, \forall k \in K \quad (5)$$

$$\sum_{i \in I} \sum_{j \in J} \sum_{k \in K} q_{ij}(Y_{iklj}^1 + Y_{iklj}^2 + Y_{iklj}^3 + Y_{ilj}) \leq Q_l, \forall l \in L \quad (6)$$

$$\sum_{k \in K} Z_k X_k \leq B \quad (7)$$

$$Y_{iklj}^1 \in \{0,1\}, Y_{iklj}^2 \in \{0,1\}, Y_{iklj}^3 \in \{0,1\}, Y_{ilj} \in \{0,1\}, Y_{ikj} \in \{0,1\} \ \forall i \in I, j \in J, k \in K, l \in L \quad (8)$$

$$X_k \in \{0,1\}, \ \forall k \in K \quad (9)$$

*3.3. Uncertain Programming Model*

3.3.1. Preliminaries of Uncertainty Theory

Uncertainty theory can offer an axiomatic system to deal with uncertain information when historical data is limited. It is different from probability theory; the experiences and judgments of field experts determine the possibility of an event happening. Moreover, compared with fuzzy set theory, uncertainty theory is more suitable for handling the transportation demand in this study because the possibility measure has no self-duality property [32]. A number of studies demonstrated that removing the self-duality property

from mathematics could produce unexpected results when modeling uncertain transportation demand [23]. We now introduce several fundamental concepts and characteristics of uncertainty theory and uncertain programming in this section, which will be used throughout the rest of the work [33].

Let $\Gamma$ be a nonempty set and $\mathcal{L}$ be a σ-algebra over $\Gamma$. Each element $\Lambda \in \mathcal{L}$ is called an event. A set function $\mathcal{M}$ from $\mathcal{L}$ to [0,1] is called an uncertain measure if it satisfies the following four axioms:

**Axiom 1** [23]. (normality axiom) $\mathcal{M}\{\Gamma\} = 1$ for the universal set $\Gamma$.

**Axiom 2** [23]. (duality axiom) $\mathcal{M}\{\Lambda\} + \mathcal{M}\{\Lambda^c\} = 1$ for any event $\Lambda$.

**Axiom 3** [23]. (subadditivity axiom) For every countable sequence of events $\{\Lambda_i\}$, we have

$$\mathcal{M}\left\{\bigcup_{i=1}^{\infty} \Lambda_i\right\} \leq \sum_{i=1}^{\infty} \mathcal{M}\{\Lambda_i\} \tag{10}$$

**Axiom 4** [23]. (product axiom) Let $(\Gamma_i, \mathcal{L}_i, \mathcal{M}_i)$ be uncertainty spaces for $i = 1,2,\dots$. Then the product uncertain measure $\mathcal{M}$ is an uncertain measure satisfying

$$\mathcal{M}\left\{\prod_{i=1}^{\infty} \Lambda_i\right\} = \bigwedge_{i=1}^{\infty} \mathcal{M}_i\{\Lambda_i\} \tag{11}$$

where $\Lambda_i$ are arbitrarily chosen events from $\mathcal{L}_i$ for $i = 1,2,\dots$, respectively.

**Definition 1** [23]. An uncertain variable $\xi$ is a measurable function from an uncertainty space $(\Gamma, \mathcal{L}, \mathcal{M})$ to the set of real numbers; that is, for any Borel set B of real numbers, the set

$$\{\xi \in B\} = \{\gamma \in \Gamma, \xi(\gamma) \in B\}$$

is an event.

For a sequence of uncertain variables $\xi_1, \xi_2, \dots, \xi_n$ and a measurable function f, Liu proved that

$$\xi = f(\xi_1, \xi_2, \dots, \xi_n)$$

defined as $\xi(\gamma) = f(\xi_1(\gamma), \xi_2(\gamma), \dots, \xi_n(\gamma)), \forall \gamma \in \Gamma$, is also an uncertain variable.

**Definition 2** [23]. The uncertainty distribution $\Phi$ of an uncertain variable $\xi$ is defined by

$$\Phi(x) = M\{\xi \leq x\}$$

for any real number x.

**Definition 3** [24]. An uncertain variable $\xi$ is called zigzag if it has a zigzag uncertainty distribution

$$\Phi(x) = \begin{cases} 0, & \text{if } x \leq a \\ \frac{x-a}{2(b-a)}, & \text{if } a \leq x \leq b \\ \frac{x+c-2b}{2(c-b)}, & \text{if } b \leq x \leq c \\ 1, & \text{if } x \geq c \end{cases} \tag{12}$$

denoted by $\mathcal{Z}$ (a,b,c) where a,b,c are real numbers with a<b<c. Figure 3 shows the Zigzag uncertainty distribution.

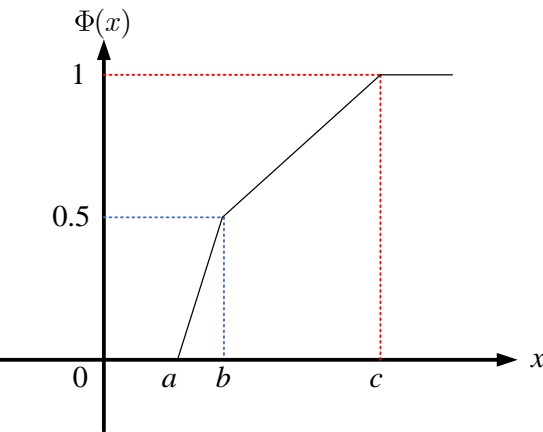

**Figure 3.** Zigzag uncertainty distribution.

The inverse uncertainty distribution of zigzag uncertain variable $\mathcal{Z}$ (a,b,c) is

$$\Phi^{-1}(\alpha) = \begin{cases} (1 - 2\alpha)\alpha + 2ab, & \text{if } \alpha < 0.5 \\ (2 - 2\alpha)b + (2\alpha - 1)c, & \text{if } \alpha \geq 0.5 \end{cases} \tag{13}$$

The zigzag uncertain variable $\xi = \mathcal{Z}$ (a,b,c) has an expected value

$$E[\xi] = \frac{a + 2b + c}{4} \tag{14}$$

**Theorem 1** [25]. *Let $\xi$ be an uncertain variable with regular uncertainty $\Phi$; if the expected value exists, then*

$$E[\xi] = \int_0^1 \Phi^{-1}(\alpha)d\alpha \tag{15}$$

*For $\xi = f(\xi_1, \xi_2, \ldots, \xi_n)$, if $\xi$ is strictly increasing functions with respect to $\xi_1, \xi_2, \ldots, \xi_n$, we have*

$$E[\xi] = \int_0^1 \Phi^{-1}(\alpha)d\alpha = \int_0^1 f(\Phi_1^{-1}(\alpha), \Phi_2^{-1}(\alpha), \ldots, \Phi_n^{-1}(\alpha))d\alpha \tag{16}$$

**Theorem 2** [25]. *Let $\xi_1, \xi_2, \ldots, \xi_n$ be independent uncertain variables with uncertainty distributions $\Phi_1, \Phi_2, \ldots, \Phi_n$, respectively. If $f(x, \xi_1, \xi_2, \ldots, \xi_n)$ and $g_i(x, \xi_1, \xi_2, \ldots, \xi_n)$ are strictly increasing functions with respect to $\xi_1, \xi_2, \ldots, \xi_n$, for $i = 1, 2, \ldots, l$, then the uncertain programming*

$$\begin{cases} \min_x E[f(x, \xi)] \\ \text{subject to} \\ \mathcal{M}\{g_i(x, \xi) \leq 0\} \geq \alpha_i, i = 1, 2, \ldots, l. \end{cases} \tag{17}$$

*is equivalent to the crisp mathematical programming*

$$\begin{cases} \min_x \int_0^1 f(x, \Phi_1^{-1}(\alpha), \Phi_2^{-1}(\alpha), \ldots \Phi_n^{-1}(\alpha))d\alpha \\ \text{subject to} \\ g_i(x, \Phi_1^{-1}(\alpha_i), \Phi_2^{-1}(\alpha_i), \ldots \Phi_n^{-1}(\alpha_i)) \leq 0, i = 1, 2, \ldots, l. \end{cases} \tag{18}$$

3.3.2. Mathematical Formulation with Uncertain Demand

The deterministic model in the previous section neglects the uncertain factors in the real-world logistics process of the cross-border multimodal container transport system.

However, under the influence of the evolving global economic situation and epidemics such as COVID-19, uncertainty has become increasingly apparent in recent years. Hence, we relax **Assumption 1** by considering the uncertain transportation demand from domestic cities to foreign hubs. In the following discussions, the demands are described as uncertain variables with a zigzag distribution function. That is, the container transportation demand between city $i$ and foreign hub $j$, which is mentioned as $q_{ij}$ in Section 3.1, can be denoted by uncertain variable $\xi_{ij}$ in this section.

In addition, the total cost has been proven in Equation (2) to be related to container transportation demand between city $i$ and foreign hub $j$. According to Definition 2, we use $\eta$ to denote the total cost, and $\eta = f(\xi_{11}, \xi_{12}, \ldots \xi_{1j}, \xi_{21}, \xi_{22}, \ldots \xi_{ij})$ is also an uncertain variable. Moreover, let the uncertainty distribution of $\xi_{ij}$ be $\Phi_{ij}$, and the uncertainty distribution of $\eta$ be $\Psi$. Given credibility confidence levels $\beta_1$ and $\beta_2$, the uncertain programming model for the problem considered in this study can be described as follows:

$$
\begin{cases}
\displaystyle \min_{x} = E\left[f(\boldsymbol{x}, \xi)\right] \\[2ex]
\text{subject to} \\[1ex]
\mathcal{M}\left\{ \displaystyle\sum_{i \in I} \sum_{j \in J} \sum_{l \in L} \xi_{ij}(Y_{iklj}^1 + Y_{iklj}^2 + Y_{iklj}^3 + Y_{ikj}) \leq Q_k^1 + X_k Q_k^2 \right\} \geq \beta_1, \ \forall k \in K \\[3ex]
\mathcal{M}\left\{ \displaystyle\sum_{i \in I} \sum_{j \in J} \sum_{k \in K} \xi_{ij}(Y_{iklj}^1 + Y_{iklj}^2 + Y_{iklj}^3 + Y_{ilj}) \leq Q_l \right\} \geq \beta_2, \ \forall l \in L \\[2ex]
\text{constrains (3) to (4), constrains (7) to (9)}
\end{cases}
\tag{19}
$$

It is clear that $f(\boldsymbol{x}, \xi_{11}, \xi_{12}, \ldots \xi_{1j}, \xi_{21}, \xi_{22}, \ldots \xi_{ij})$ and $g_k(\boldsymbol{x}, \xi_{11}, \xi_{12}, \ldots \xi_{1j}, \xi_{21}, \xi_{22}, \ldots \xi_{ij})$ are strictly increasing functions with respect to $\xi_{11}, \xi_{12}, \ldots \xi_{1j}, \xi_{21}, \xi_{22}, \ldots \xi_{ij}$, for $i = 1,2, \ldots ,r$, $j = 1,2, \ldots ,s$. Therefore, according to **Theorem 1** and **Theorem 2**, the uncertain program is equivalent to the crisp mathematical model:

$$
\begin{cases}
\min = C_1\big(E(q_{ij})\big) + C_2\big(E(q_{ij})\big) + C_3\big(E(q_{ij})\big) + C_4\big(E(q_{ij})\big) + C_5 \\[1ex]
\text{where } C_1(\cdot) - C_4(\cdot) \text{ and } C_5 \text{ are explained in equation (2)} \\[1ex]
\text{subject to} \\[1ex]
\displaystyle\sum_{i \in I} \sum_{j \in J} \sum_{l \in L} \Phi_{ij}^{-1}(\beta_1)(Y_{iklj}^1 + Y_{iklj}^2 + Y_{iklj}^3 + Y_{ikj}) \leq Q_k^1 + X_k Q_k^2, \forall k \in K \\[3ex]
\displaystyle\sum_{i \in I} \sum_{j \in J} \sum_{k \in K} \Phi_{ij}^{-1}(\beta_2)(Y_{iklj}^1 + Y_{iklj}^2 + Y_{iklj}^3 + Y_{ilj}) \leq Q_l, \forall l \in L \\[2ex]
\text{constrains (3) to (4), constrains (7) to (9)}
\end{cases}
\tag{20}
$$

Apparently, the program (20) is a binary integer programming model. The independent variable in program (20) is $X_k$ and $Y_{iklj}^1, Y_{iklj}^2, Y_{iklj}^3, Y_{ilj}, Y_{ikj}$, and the objective is to find a network solution including nodes and routes to minimize the total cost under different scenarios. Program (20) can be directly solved by general optimizers that are suitable for binary integer programming models.

## 4. Case Study

To show the applicability of the proposed models, we conducted a case study concerning the Huaihai Economic Zone-Europe multimodal container transport system. All the experiments were implemented on a Lenovo Thinkpad laptop with an Intel(R) Core (TM) i7-1165G7@2.80 GHz CPU and 16.00 GB RAM using Microsoft Windows 10 (64-bit). All optimization models were solved directly by applying Lingo 18.0 with default settings.

*4.1. Description of the Huaihai Economic Zone—Europe Multimodal Container Transport System*

4.1.1. Introduction of Huaihai Economic Zone

The Huaihai Economic Zone is situated in the eastern bridgehead area of the Asia-Europe Continental Bridge, bordered by the Zhongyuan Economic Zone to the west, the

Yangtze River Delta Economic Zone to the south, and the Bohai Economic Zone to the north [34,35]. As illustrated in Figure 4, due to its prominent location at the eastern bridgehead of the Asia-Europe Continental Bridge, it has assumed the essential role of a relay station that serves the east-west convergence of China's economy.

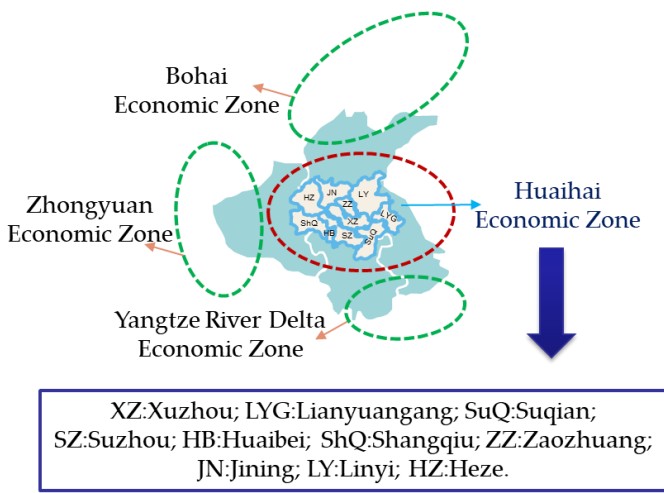

**Figure 4.** Location of Huaihai Economic Zone.

The concept of the Huaihai Economic Zone was first proposed in 1986, and the broad Huaihai Economic Zone includes 20 cities in four provinces: Jiangsu, Shandong, Henan, and Anhui. In 2018, the National Development and Reform Commission of China [34] clarified the exact scope of the Huaihai Economic Zone, which includes a total of ten cities, as listed in Figure 4.

### 4.1.2. Huaihai Economic Zone—Europe Multimodal Container Transport System

The Huaihai Economic Zone-Europe multimodal container transport system in this study contains four types, 21 nodes in total. For origin cities, we considered all cities in the Huaihai Economic Zone, as listed in Figure 4. For logistics parks, we selected six nodes in the Huaihai Economic Zone, Xuzhou logistics park (XZLP): Suzhou logistics park (SZLP), Yudong logistics park (YDLP), Linyi logistics park (LYLP), Yanzhou logistics park (YZLP), and Zaozhuang logistics park (ZZLP). For seaports, we selected Lianyungang port and Rizhao port. It is important to note that the Rizhao port, which is located in the broad Huaihai Economic Zone, has been included in the network in order to meet the diversity of node selection. For foreign hubs, we selected three hubs that have both seaport and inland port functions: Rotterdam, Hamburg, and Duisburg.

### 4.1.3. Data Collection

In this section, we specify the relevant constant data for the case study. According to [36], combined with Equation (1), Table 2 reports the transportation costs and carbon emissions under different transportation modes. Table 3 shows the average annual investment for inland port conversion and the annual freight handling capability before/after conversion. The data source for this table is based on development plans for each inland port [35]. It should be noted that the useful life of inland ports is calculated on a 50-year basis, and cross-border services account for approximately 20% of the overall investment in the conversion of inland ports.

**Table 2.** Transportation costs and carbon emissions under different transportation modes.

| Transportation Mode | Road | Railway | Inland Water Transportation | Shipping | CRE |
|---|---|---|---|---|---|
| Transportation cost [1] (yuan/TEU·km) | 10 | 2.7 | 1.0 | 1.5 | 2.5 |
| Carbon emission [1] (tonne/TEU·km) | $1.77 \times 10^{-3}$ | $9.0 \times 10^{-4}$ | $3.0 \times 10^{-4}$ | $3.0 \times 10^{-4}$ | $8.0 \times 10^{-4}$ |

[1] Data based on [31,36,37].

Additionally, Table 3 also shows the capacity of domestic seaports and foreign hubs. Table 4 uses historical data provided by [38] as the basis for statistics and analysis of the relevant levels of demand to provide a rough measure of cross-border transportation demand between each city in the Huaihai Economic Zone and each foreign hub. For clarity of presentation, the transport distance between different nodes by different transport modes is relegated to Appendix A in this paper.

**Table 3.** Average annual investment for conversion and capacity before/after conversion.

| Node | XZLP | SZLP | YDLP | LYLP | YZLP | ZZLP | LYG Port | RZ Port | Rotterdam Port | Hamburg Port | Duisburg Port |
|---|---|---|---|---|---|---|---|---|---|---|---|
| Origin Capacity (10,000TEU) | 10 | 5 | 5 | 10 | 5 | 10 | 100 | 100 | 300 | 300 | 300 |
| Capacity Added by Conversion (10,000TEU) | 50 | 5 | 10 | 20 | 15 | 30 | - | - | - | - | - |
| Average annual investment for conversion (10,000 yuan) | 2000 | 500 | 1000 | 1000 | 1000 | 1500 | - | - | - | - | - |

**Table 4.** Cross-border transportation demand for different cities in Huaihai Economic Zone.

| Unit: 10,000TEU | Rotterdam | Hamburg | Duisburg |
|---|---|---|---|
| Xuzhou | 20 | 5 | 5 |
| Lianyungang | 5 | 5 | 2 |
| Suqian | 5 | 3 | 2 |
| Suzhou | 5 | 2 | 3 |
| Huaibei | 5 | 2 | 5 |
| Shangqiu | 3 | 4 | 2 |
| Zaozhuang | 5 | 5 | 5 |
| Jining | 5 | 10 | 5 |
| Linyi | 10 | 10 | 5 |
| Heze | 5 | 5 | 10 |

*4.2. Computational Results for Deterministic Model*

We first conduct a case study to validate the efficiency of the deterministic model. Figure 5 shows the network structure when the transportation cost of CRE is 2.0 or 2.8. It appears that the location of inland ports and the transportation route will change significantly when the transportation cost of CRE differs. More specifically, when the transportation cost of CRE is 2.0, four logistics parks (XZLP, SZLP, LYLP, and ZZLP) are converted into THE inland ports, and eight CRE routes are selected. However, when the transportation cost of CRE is 2.8, only XZLP and ZZLP are expanded into inland ports, and only four CRE routes are selected.

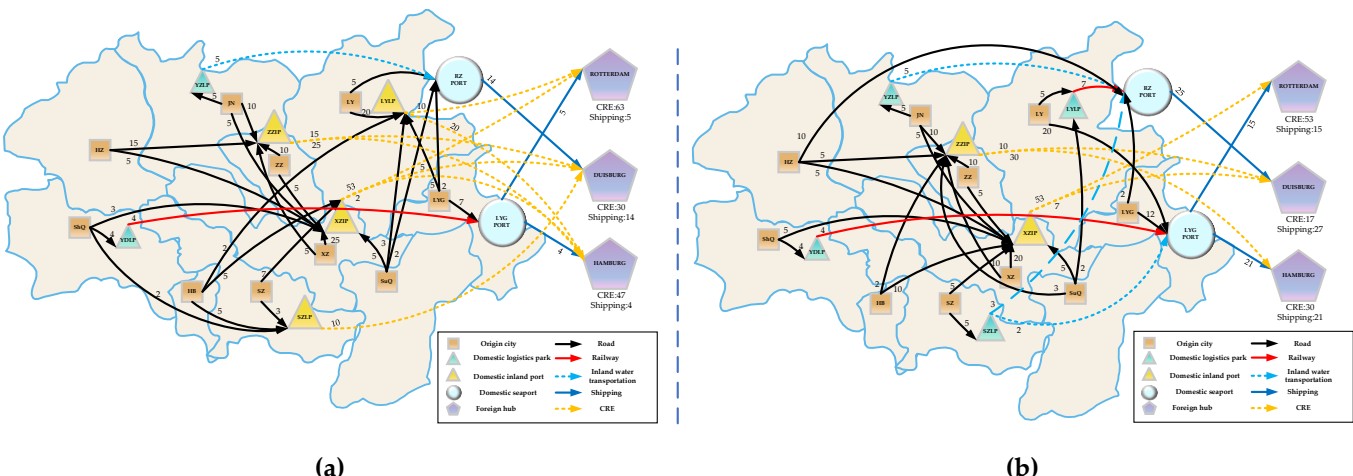

**(a)**                                                                                      **(b)**

**Figure 5.** (**a**) Network layout when transportation cost of CRE is 2.0; (**b**) network layout when transportation cost of CRE is 2.8.

Note that the capacity of inland ports is also one of the most important factors influencing the layout of the network. The estimated capacity of the inland port in this paper refers to the capacity in the planning scenario. In practice, due to the phasing of inland port construction and subsequent expansion, the actual increased capacity of the logistics park after expansion into an inland port may also change. As can be seen in Figure 6, the proportion of CRE grows with inland port capacity, and the proportion of CRE decreases significantly when the actual added capacity at inland ports is less than the estimated added capacity.

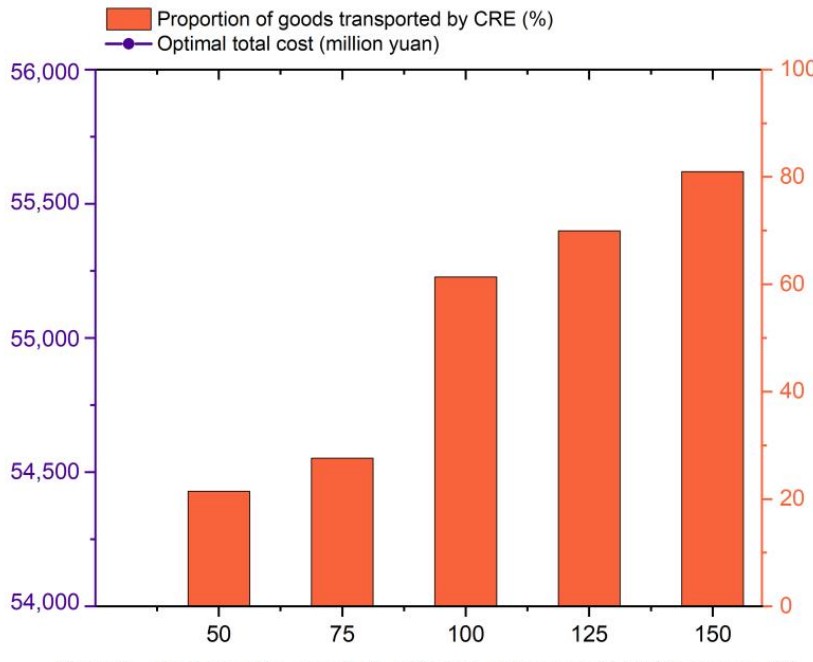

**Figure 6.** Changes in the proportion of CRE and total optimal costs when actual added capacity are changed.

Figure 7 suggests that the optimal total cost of the system increases as the CRE cost rises, while the total cost under no investment restrictions is lower when the CRE cost is less than 2.4. The total cost with no investment restrictions would also be lower when the CRE cost is less than 2.4. Moreover, as the transportation cost of CRE decreases, the

proportion of CRE carried tends to increase progressively. The increase in the proportion of CRE with no investment restrictions is particularly noticeable when the CRE cost is less than 2.4. Figure 8 reveals that the optimal total cost of the system is also affected by the carbon emissions of CRE and the value of the carbon tax. There is evidence to indicate that the rise in carbon emissions of CRE and carbon tax gives rise to the total cost.

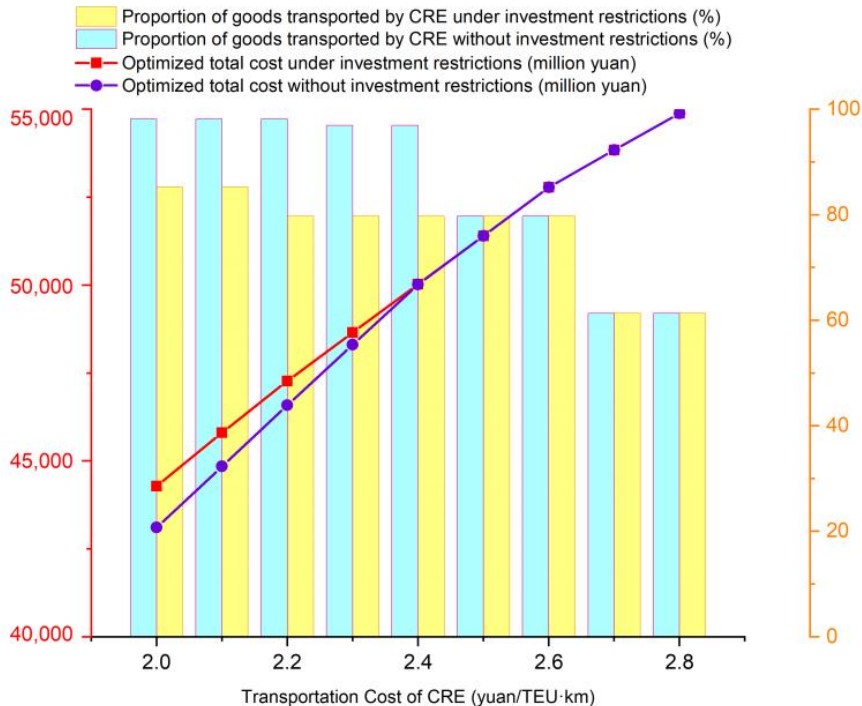

**Figure 7.** Changes in the proportion of CRE and total optimization costs when CRE transport costs and investment limits are changed.

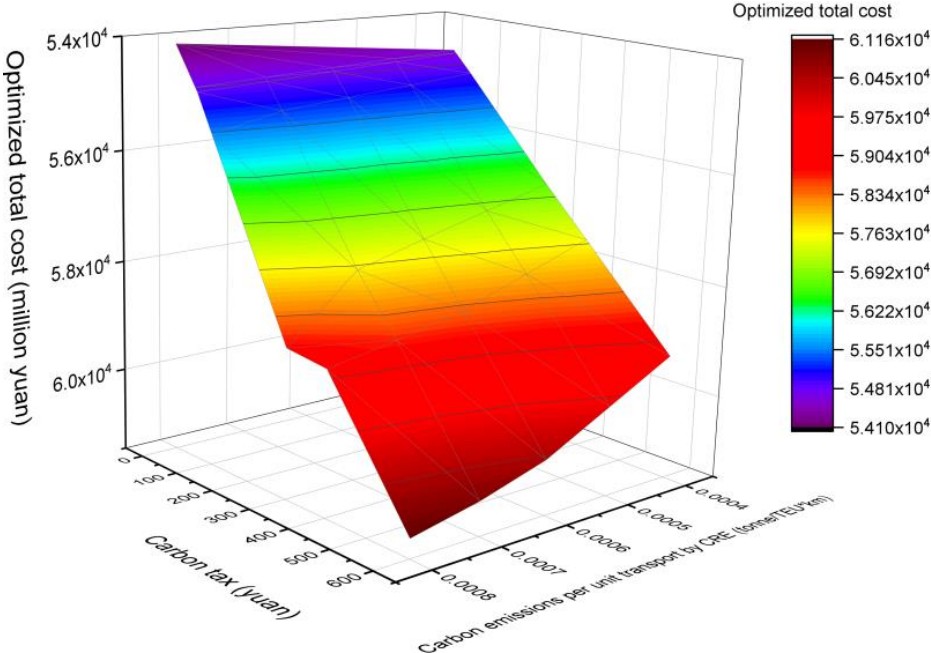

**Figure 8.** Changes in optimal costs when the carbon emissions of CRE and carbon tax change.

### 4.3. Computational Results for Uncertain Model

We then implement the uncertain programming model by adopting the uncertainty demand in Appendix B. It can be observed from Figure 9 that the optimal total cost rises when $\beta_1$ and $\beta_2$ rise. That is to say, in order to satisfy greater uncertainty, the total cost needs to be sacrificed. This result confirmed for us that the optimization process under uncertainty needed to be reconsidered, and we therefore carried out a further detailed analysis of the situation relevant when the parameters are varied under uncertainty.

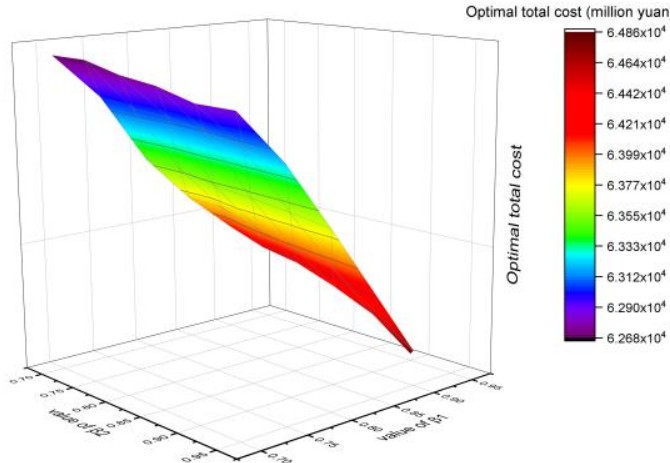

**Figure 9.** Optimal total cost versus values of $\beta_1$ and $\beta_2$.

It can be seen from Figure 10 that the total carbon emissions are influenced synergistically by the parameters $\beta_1$ and $\beta_2$. Note that the two extreme values occur when $\beta_1 = 0.95$, $\beta_2 = 0.7$, and $\beta_1 = 0.95$, $\beta_2 = 0.80$. It is obvious that the carbon emission from shipping and inland water transportation is much lower than other transportation modes. Generally speaking, the proportion of shipping rises when $\beta_1$ increases and $\beta_2$ decreases, and when $\beta_1 = 0.95$, in order to meet the capacity constraints for inland ports, more goods are transported by shipping instead of CRE, which can reduce the total carbon emission. However, since the objective of the model is to minimize the expected value of the total cost of the system, sometimes the route with a lower cost and higher carbon emission will also be chosen, which explains why in the scenario $\beta_1 = 0.95$, $\beta_2 = 0.75$, the carbon emission is higher than in the scenario $\beta_1 = 0.95$, $\beta_2 = 0.8$.

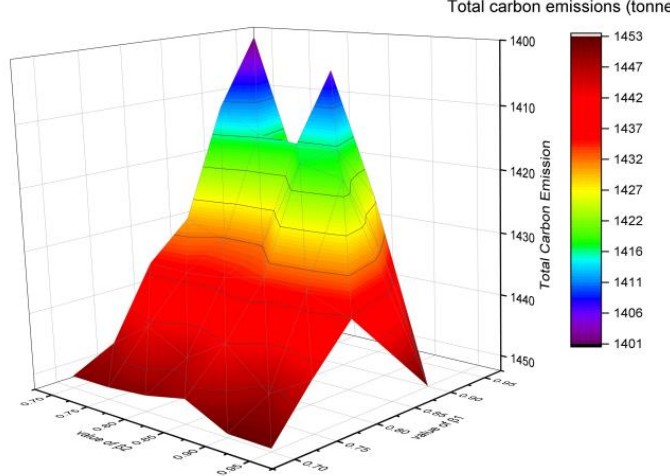

**Figure 10.** Carbon emissions versus values of $\beta_1$ and $\beta_2$.

Figure 11 depicts the changes in the volume of goods transported by CRE when $\beta_1$, $\beta_2$, and the capacity of inland ports change. Firstly, it can be found that when the capacity of the inland port is deterministic, the larger $\beta_1$ is, the less volume is generally transported by CRE. In other words, when the constraint on the logistics park and inland port capacity is tightened, shipping will be more likely to be chosen as the transport mode linking the Huaihai Economic Zone to Europe. However, it can be noted that there is a set of anomalous data in the results: the proportion of CRE is higher when $\beta_1 = 0.95$ and proportion = 75% than when $\beta_1 = 0.90$ and ratio = 75%. This is because the inland port option when $\beta_1 = 0.90$ and ratio = 75% cannot meet the capacity constraint when $\beta_1 = 0.95$ and ratio = 75%. By replacing Yanzhou inland port with Linyi inland port, the overall capacity of the inland port increased, and the optimal routes under this scenario, by increasing the relative total cost, gave rise to the volume of goods transported by CRE. Moreover, the evidence in Figure 11 also suggests that when other factors are determined, the value of $\beta_2$ does not have a major impact on the choice of inland ports and the proportion of CRE, and only changes to several individual routes will be made.

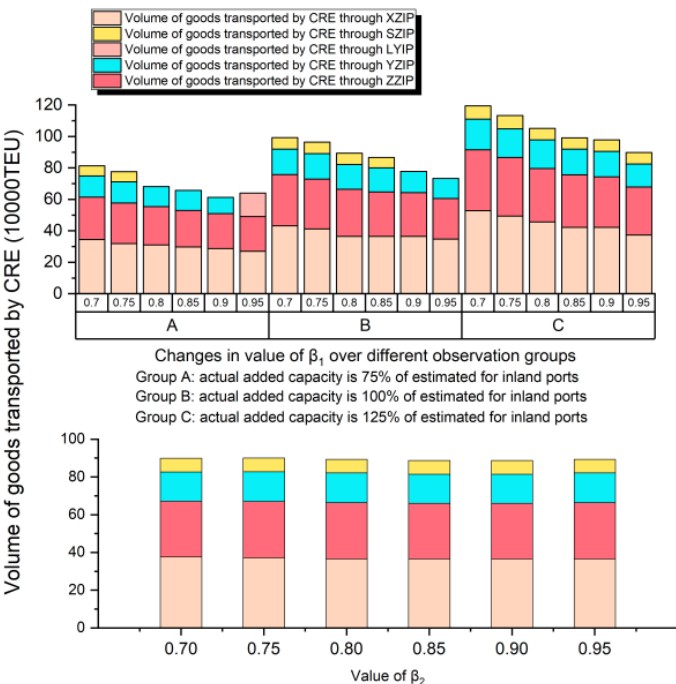

**Figure 11.** Changes in volume of goods transported by CRE when β1, β2, and capacity of inland ports changes.

Apparently, according to Figure 12, the proportion of goods transported by CRE under uncertain demand shows a significant decrease compared with the situation under deterministic demand. In addition, when $\beta_1$ is determined, the increase in inland port capacity will increase the proportion of goods transported by CRE.

Table 5 shows the inland port locations, CRE routes, and shipping routes of different plans. The optimal network plan for the deterministic demand case, called Plan A, is not feasible under an uncertain demand situation. Under plan A, the estimated volume of goods transported through the Xuzhou inland port is 728,500 TEU, which is beyond the capacity of the inland port. To better describe the actual problem, four optimal network plans were made for the solution under uncertainty, taking into account the different values of the two parameters, as shown in Figure 13.

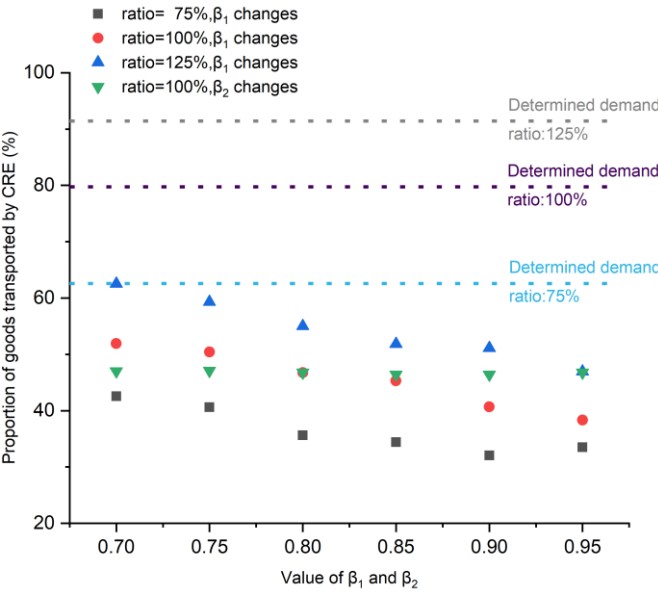

**Figure 12.** Changes in proportion of goods transported by CRE when $\beta_1$, $\beta_2$, and capacity of inland ports changes.

**Table 5.** Computational results in Plan A to Plan E.

| | Inland Port Locations | CRE Routes | Shipping Routes |
|---|---|---|---|
| Plan A: Deterministic demand situation | XZ, SZ, YZ, ZZ | XZ inland port–Rotterdam XZ inland port–Duisburg SZ inland port–Duisburg YZ inland port–Duisburg ZZ inland port–Hamburg | RZ port–Duisburg LYG port–Rotterdam LYG port–Hamburg |
| Plan B: Uncertain demand situation, $\beta_1 = 0.8$, $\beta_2 = 0.8$ | XZ, SZ, YZ, ZZ | XZ inland port–Rotterdam SZ inland port–Duisburg YZ inland port–Rotterdam YZ inland port–Duisburg ZZ inland port–Duisburg ZZ inland port–Hamburg | RZ port–Duisburg RZ port–Rotterdam LYG port–Hamburg LYG port–Rotterdam |
| Plan C: Uncertain demand situation, $\beta_1 = 0.9$, $\beta_2 = 0.8$ | XZ, YZ, ZZ | XZ inland port–Rotterdam XZ inland port–Duisburg XZ inland port–Hamburg YZ inland port–Duisburg ZZ inland port–Duisburg ZZ inland port–Hamburg | RZ port–Duisburg RZ port–Rotterdam LYG port–Hamburg LYG port–Rotterdam |
| Plan D: Uncertain demand situation, $\beta_1 = 0.8$, $\beta_2 = 0.9$ | XZ, SZ, YZ, ZZ | XZ inland port–Rotterdam SZ inland port–Duisburg YZ inland port–Hamburg YZ inland port–Duisburg ZZ inland port–Hamburg | RZ port–Duisburg RZ port–Rotterdam LYG port–Hamburg LYG port–Rotterdam |
| Plan E: Uncertain demand situation, $\beta_1 = 0.9$, $\beta_2 = 0.9$ | XZ, SZ, YZ, ZZ | XZ inland port–Rotterdam XZ inland port–Duisburg SZ inland port–Duisburg YZ inland port–Duisburg ZZ inland port–Hamburg | RZ port–Duisburg RZ port–Rotterdam LYG port–Hamburg LYG port–Rotterdam |

As illustrated in Table 5 and Figure 13, the optimal network plan changes a lot in different scenarios. However, the inland port locations do not change a lot, except for Plan C. This may imply that when other factors are determined, the inland port locations will not change significantly due to the uncertainty of demand. Moreover, compared with Plan

A, Plans B to E all add a shipping route from Rizhao Port to Rotterdam. It can be concluded that shipping is more stable than CRE under uncertain demand.

**Figure 13.** Optimal network plans on different scenarios.

## 5. Discussions and Policy Guidance

Our research provides the following discussions and policy guidance according to the results of the numerical experiments:

### 5.1. Inland Port Selections

- **The selection plan for inland ports is robust.** Our results demonstrated that the selected plan of inland ports is robust against uncertain demand. Table 5 and Figure 13 show that four of the five plans choose XZIP, SZIP, YZIP, and ZZIP. Additionally, investment in inland ports is a long-term process with multiple investment risks, and the findings of this paper provide some support for investment decisions in inland ports. That is, when the investment and construction of an inland port is a choice made after scientific analysis, the inland port will have a certain degree of robustness and will be able to cope with the situation under changing demand.

- **There should not be too many inland ports in a certain region.** According to the results, none of the plans converts all six logistics parks into inland ports. It is reasonable to assume that the number of inland ports in a certain region should be limited. Otherwise, although some inland ports are built, they are not selected in the optimal network plan, which means the capital resources for construction will be tied up while the total cost of the whole system will increase.

- **The construction of inland ports has priority.** As indicated in Figure 11, goods transported through XZIP and ZZIP by CRE account for 75% of the total goods transported by CRE, which suggests that the construction of these two inland ports

has a high impact on the optimal solution for the whole network and should be first considered to be built when the investment amount is not sufficient to build all four inland ports. The conclusion is consistent with the real situation that XZIP and ZZIP are built now, especially for XZIP, which has run over 1000 CRE trains in 2021 [3,5].

### 5.2. Transportation Routes Choices

- **CRE offers advantages in reducing total cost.** As illustrated in Figures 5 and 6, the rise in the proportion of goods transported by CRE decreases the total cost. We can assume that, although the transport cost per unit transported by shipping is lower than by CRE, the distance by shipping is longer, and the transfer time is increasing, it is likely that in most of the cross-border transport processes, CRE is more cost-effective than shipping. In the actual transport process, it will generally take 30–48 days for goods to be transported from China to Europe by shipping, but only 20–25 days by CRE [5]. The result is also consistent with the fact that CRE will save nearly 8–20% of the total cost compared with shipping [5].
- **Shipping is robust against uncertain demand.** According to Table 5 and Figure 13, it is likely to be assumed that shipping can satisfy a higher credibility confidence level against uncertain demand. A plausible explanation for this phenomenon can be attributed to the fact that seaports generally have a greater handling capacity than inland ports.
- **Road is the main transportation mode for short distances within the country.** Combining Figures 5 and 13 and Table 5, there is evidence to prove that road is still the main mode of transportation within the Huaihai Economic Zone. According to Tables A1 and A2, the distances between nodes within the Huaihai Economic Zone are less than 500 km. Under all scenarios, the road is generally the primary mode of transport in the case we study.

### 5.3. Strategies for Reducing the Total Cost

- **Reduce transport costs for CRE.** Results in Figures 6 and 7 support the opinion that the total cost increases as the transport cost of CRE increases, and the lower transportation cost of CRE gives rise to the proportion of goods transported by CRE. In the actual operation process, we can reduce the transport cost of CRE by increasing the full load rate, innovating the organization of CRE trains, and making technical innovations to the CRE carriers. Moreover, the marginal effect is larger when the cost of the CRE is 2.1 as well as 2.6. Thus, when the transport cost of CRE is a little over this value, we can also consider using subsidies to achieve marginal benefits.
- **Reduce carbon emissions from CRE.** Figure 8 proves the view that the lower carbon emissions of CRE and lower carbon tax help to reduce the total costs. The carbon tax in China is CNY 54.22 on average in 2021 [39], and when the carbon tax is CNY 600 (which is similar to the average carbon tax in Europe), the slope is large, which means that the carbon emissions of CRE influence the total cost a lot in this scenario. The evidence suggests reducing the carbon emissions of CRE by, for example, improving energy conversion rates to adapt to an increasing carbon tax in the future.
- **Expand investment limits under suitable conditions.** Figure 7 shows that when the transportation cost of CRE is less than 2.4, it is advisable to expand investment limits. The primary way to raise the investment limit is to optimize the structure of the investors. The current main investors in the inland port are mainly the local government, but in the future, the role of logistics real estate developers in inland port investment can be fully exploited.

### 5.4. Schemes for Improving Network Performance against Uncertain Demand

- **Increase the capacity of inland ports.** Figures 11 and 12 provide evidence that, under an uncertain demand situation, the change in $\beta_1$ has a greater impact on the network optimal plan compared with the change in $\beta_2$. And Figure 11 also demonstrates that

when inland port capacity is increased to 125%, the network plan is more stable under different changes of $\beta_1$, the inland port location plan is not changing, and the route selection does not change a lot. In order to get a higher credibility confidence level, the inland port capacity should be increased as much as possible.

- **Add suitable shipping routes to the network.** Figure 13 and Table 5 support the idea that by adding a new shipping route to the network, the network's robustness under uncertain demand is improved. It can be assumed that the shipping mode can help the whole system to withstand the risk that the logistics network is not able to meet transport demand due to uncertainty.

## 6. Conclusions

In this paper, we investigated a cross-border multimodal container transport system based on inland ports. We formulated both deterministic and uncertain models for the cross-border multimodal container transportation network design problem. Based on uncertainty theory, we converted the uncertain model into a crisp model. We provided a case study of the Huaihai Economic Zone-Europe multimodal container transport system. The results concluded the suggestions and policy guidance over four issues: the inland port locations, the transportation route choices, the strategies to reduce the total cost, and the schemes to improve network performance against uncertain demand.

The novelty of this work can be summarized as follows:

- It is distinct from the aforementioned work in that we focus on the important role of inland ports in the cross-border container transportation system, particularly considering CRE as a significant transportation mode.
- It is the first work in which the uncertain transportation demand, carbon emission, and customs clearance cost are jointly considered to determine the inland port location as well as the cross-border transportation routes and modes. Based on this, the obtained results can provide strong practical guidance.
- This work presents an integrated uncertain programming model by combining the expected value model and a chance-constrained program to formulate the cross-border multimodal container transportation network design problem under uncertain transportation demand.

It should be noted that the carbon emissions of different modes of transportation are changing greatly due to the application of new transportation carriers such as new energy vehicles and LNG ships. However, the above-mentioned situation is not considered in this paper. Hence, in future work, we will address the cost of carbon emissions in a more precise way. Furthermore, in this paper, we have found that the objectives of minimizing total costs and minimizing carbon emissions are not always consistent. Therefore, we will obtain the optimal plan under multiple objectives in the future.

**Author Contributions:** All authors contributed to the study conception and design. Methodology and software, J.M. and K.Y.; formal analysis and investigation, J.M., K.Y. and L.J.; data curation and writing—original draft preparation, J.M.; writing—review and editing, X.W. and K.Y; supervision, X.W.; funding acquisition, X.W. All authors have read and agreed to the published version of the manuscript.

**Funding:** This research was supported by the Major Scientific and Technological Innovation Project of Shandong Province (No. 2019JZZY020715).

**Data Availability Statement:** Not applicable.

**Conflicts of Interest:** The authors declare no conflict of interest.

## Appendix A. Transport Distance between Different Nodes by Different Modes

**Table A1.** Transport distance from origin city to domestic logistics parks/domestic seaports by road.

| Unit: km | XZLP | SZLP | YDLP | LYLP | YZLP | ZZLP | LYG PORT | RZ PORT |
|---|---|---|---|---|---|---|---|---|
| Xuzhou | 20 | 100 | 160 | 200 | 180 | 75 | 230 | 308 |
| Lianyungang | 225 | 300 | 400 | 150 | 260 | 225 | 20 | 130 |
| Suqian | 125 | 200 | 270 | 160 | 300 | 190 | 170 | 250 |
| Suzhou | 80 | 10 | 190 | 300 | 290 | 180 | 320 | 400 |
| Huaibei | 60 | 50 | 160 | 260 | 260 | 150 | 287 | 370 |
| Shangqiu | 170 | 120 | 15 | 370 | 240 | 250 | 394 | 470 |
| Zaozhuang | 90 | 170 | 250 | 144 | 125 | 15 | 240 | 265 |
| Jining | 160 | 255 | 230 | 205 | 40 | 152 | 340 | 301 |
| Linyi | 210 | 275 | 360 | 20 | 190 | 150 | 125 | 160 |
| Heze | 280 | 220 | 130 | 310 | 140 | 270 | 500 | 405 |

**Table A2.** Transport distance from domestic logistics park to domestic seaports by different modes.

| Unit: km | Road | | Railway | | Inland Waterway | |
|---|---|---|---|---|---|---|
| | LYG Port | RZ Port | LYG Port | RZ Port | LYG Port | RZ Port |
| XZLP | 250 | 320 | 185 | 300 | 500 | 700 |
| SZLP | 320 | 400 | 300 | 350 | 400 | 800 |
| YDLP | 400 | 480 | 350 | 420 | -[1] | - |
| LYLP | 200 | 170 | 180 | 150 | - | - |
| YZLP | 310 | 270 | 400 | 295 | 450 | 400 |
| ZZLP | 250 | 270 | 350 | 300 | 500 | 400 |

[1] The two nodes cannot be linked by inland waterway

**Table A3.** Transport distance from seaports to foreign hubs by shipping.

| Unit: km | Rotterdam | Hamburg | Duisburg |
|---|---|---|---|
| LYG Port | 20,822 | 20,224 | 21,851 |
| RZ Port | 21,520 | 21,396 | 20,056 |

**Table A4.** Transport distance from inland ports to foreign hubs by CRE.

| Unit: km | Rotterdam | Hamburg | Duisburg |
|---|---|---|---|
| XZLP | 10,400 | 11,683 | 11,000 |
| SZLP | 12,010 | 12,200 | 10,855 |
| YDLP | 12,900 | 12,350 | 12,860 |
| LYLP | 12,300 | 11,800 | 13,400 |
| YZLP | 12,100 | 12,000 | 11,050 |
| ZZLP | 12,780 | 10,430 | 10,890 |

## Appendix B. Cross-Border Transportation Uncertain Demand between Different Origin Cities and Foreign Hubs

**Table A5.** Cross-border transportation uncertain demand between origin cities and Rotterdam.

| Unit: 10,000TEU | Rotterdam (a) [1] | Rotterdam (b) [1] | Rotterdam (c) [1] |
| --- | --- | --- | --- |
| Xuzhou | 5 | 20 | 40 |
| Lianyungang | 2 | 5 | 30 |
| Suqian | 2 | 5 | 10 |
| Suzhou | 0.5 | 5 | 20 |
| Huaibei | 0.5 | 5 | 20 |
| Shangqiu | 1 | 3 | 5 |
| Zaozhuang | 2 | 5 | 20 |
| Jining | 3 | 5 | 20 |
| Linyi | 5 | 10 | 20 |
| Heze | 2 | 5 | 15 |

[1] (a) refers to the minimum demand between nodes; (b) refers to the most likely demand between nodes; (c) refers to the maximum demand between nodes

**Table A6.** Cross-border transportation uncertain demand between origin cities and Hamburg.

| Unit: 10,000TEU | Hamburg (a) | Hamburg (b) | Hamburg (c) |
| --- | --- | --- | --- |
| Xuzhou | 3 | 5 | 10 |
| Lianyungang | 3 | 5 | 8 |
| Suqian | 1 | 3 | 5 |
| Suzhou | 1 | 2 | 5 |
| Huaibei | 1 | 2 | 4 |
| Shangqiu | 2 | 4 | 5 |
| Zaozhuang | 3 | 5 | 8 |
| Jining | 5 | 10 | 15 |
| Linyi | 8 | 10 | 20 |
| Heze | 2 | 5 | 7 |

**Table A7.** Cross-border transportation uncertain demand between origin cities and Duisburg.

| Unit: 10,000TEU | Duisburg (a) | Duisburg (b) | Duisburg (c) |
| --- | --- | --- | --- |
| Xuzhou | 3 | 5 | 10 |
| Lianyungang | 1 | 2 | 5 |
| Suqian | 0.5 | 2 | 3 |
| Suzhou | 1 | 3 | 4 |
| Huaibei | 1 | 5 | 7 |
| Shangqiu | 1 | 2 | 3 |
| Zaozhuang | 2 | 5 | 10 |
| Jining | 2 | 5 | 10 |
| Linyi | 3 | 5 | 10 |
| Heze | 8 | 10 | 15 |

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
