# Peer review of "Uncertain Programming Model for the Cross-Border Multimodal Container Transport System Based on Inland Ports"

_axioms, doi:10.3390/axioms12020132_

Round 1
Reviewer 1 Report
The manuscript presented to me for review is exemplary. It contains essential elements that should be included in every article.
In the introduction, the authors presented the research problem and correctly formulated 3 research questions. They defined what techniques they would use to solve the research problem, what limitations they entail, and what is the purpose of the research. They also provided a brief description of the structure of the article.
The review of the literature is sufficient, although it lacks many scientifically valuable publications and I regret that the authors did not reach them.
The presented research methodology is reader-friendly, and clear and does not contain unnecessary frills.
Formula 1 is incorrectly written.
For some other designs, there is no description of the markings.
I have no comments on the empirical part. In my opinion, it is properly described, and balanced. The language of conclusions is addressed to everyone, not just experts in quantitative methods. The interpretations are correct and complete.
The discussion brings cognitive value. Does not require refilling.
In the conclusions, the authors should add their novelty, the research results' usefulness, and limitations.
Maybe the authors will find inspiration in:
1. Szaruga, E.; Kłos-Adamkiewicz, Z.; Gozdek, A.; Załoga, E. Linkages between Energy Delivery and Economic Growth from the Point of View of Sustainable Development and Seaports. Energies 2021, 14, 4255. https://doi.org/10.3390/en14144255
2. Bernacki, D.; Lis, C. Investigating the Future Dynamics of Multi-Port Systems: The Case of Poland and the Rhine–Scheldt Delta Region. Energies 2022, 15, 6614. https://doi.org/10.3390/en15186614
3. Kotowska, I.; Mańkowska, M.; Pluciński, M. Inland Shipping to Serve the Hinterland: The Challenge for Seaport Authorities. Sustainability 2018, 10, 3468. https://doi.org/10.3390/su10103468
Author Response
Dear Editor and Reviewer,
Thanks very much for taking your time to give constructive comments on our manuscript. We have carefully considered the suggestion from you and made extensive corrections to our previous work. For the point-by-point response, please see the attachment.
Best regards,
Authors

Reviewer 2 Report
The present manuscript is focus on a uncertain programming model for the cross-border multi-modal container transportation network design problem. To this matter, authors consider the concept of uncertainty mainly based on [20-22].
As curiosity, on page 7, at the bottom, Authors write that "In contrast to probability theory, the expert opinion determines the occurrence possibility of an event happening". These is a classic concept in fuzzy sets, what is used to model uncertainty. However, no discussion about it is offered, why?
The programming model under uncertainty is offered in 3.3. In 3.3.1, we find the definitions, axioms, theorems,.. all referenced to [20-22]. To this latter, I miss some (at least one) original contribution. In 3.3.2, authors include the formulation based on the latters. Section 4 is the application case, and it is the main contribution.
In my opinion, only a short novelty contribution of this manuscript is only in Section 4, and it is related to modelling.
Author Response

(The authors gave the same response as above.)

Reviewer 3 Report
Dear editor and authors,
The manuscript is generally important, well-written and well-presented. The results look clear and correct. Also, the presentation of the manuscript is well-written. This manuscript has both theoretical results and applications. In addition, the manuscript includes figures, diagrams and it compares the relations with the previous one. Moreover, the mathematical background of this manuscript is sufficient for the publication. English language should be improved. In view of the above discussions, it is my opinion that the manuscript should be accepted and it is qualified to be published in "Axioms".
Regards
Author Response

(The authors gave the same response as above.)

Round 2
Reviewer 2 Report
Authors have revised the current version, with the inclusion of my suggestions.